# Clinical value of Lipoprotein(a) combined with CatLet coronary score in predicting adverse events after emergency PCI for AMI patients

Mengru Wang[1], Fudong Hu[2], Rongyan Jiang[1], Sheng Tu[1]*

1 Department of Cardiology, Bozhou Hospital Affiliated to Anhui Medical University, Bozhou City, Anhui Province, China, 2 Department of Cardiology, Department of Cardiology, The First Affiliated Hospital of Zhengzhou University, Zhengzhou, Henan, People's Republic of China

* tusheng@ahmu.edu.cn

## Abstract

### Background and aims

Lipoprotein(a) [Lp(a)] promotes atherosclerotic plaque vulnerability through pro-inflammatory and thrombogenic pathways, while the CatLet© angiographic score quantifies coronary lesion complexity. We hypothesized that their integration would improve prognostication in acute myocardial infarction (AMI) after emergency percutaneous coronary intervention (ePCI).

### Methods

In this retrospective cohort, 307 AMI patients undergoing successful ePCI (2020–2022) were stratified by 1-year major adverse cardiovascular/cerebrovascular events (MACCE). Serum Lp(a) and troponin I were measured post-admission. CatLet© and Gensini scores were assessed by blinded analysts. Multivariable logistic regression and ROC analyses evaluated predictive performance.

### Results

MACCE patients (n = 78) exhibited higher Lp(a) (135.99 ± 33.07 vs. 123.35 ± 42.70 nmol/L, P = 0.0178) and CatLet© scores (33.58 ± 9.04 vs. 30.80 ± 8.24, P = 0.0012) versus controls. Lp(a) (OR=2.339, 95%CI:1.519–3.603, P < 0.001) and CatLet© score (OR=1.092, 95%CI:1.027–1.161, P = 0.005) independently predicted MACCE. The combined model Lp(a)≥70.70 nmol/L + CatLet© ≥ 18.6) significantly outperformed individual markers (AUC 0.862 [95%CI:0.83–0.96] vs. 0.780/0.833; DeLong's test confirmed the superiority of the combined model over individual predictors (P = 0.0089, Z = 2.64 vs. Lp(a); P = 0.034, Z = 2.12 vs. CatLet© score), with 88% sensitivity and 83% specificity.

**Data availability statement:** All relevant data are within the manuscript and its Supporting Information files.

**Funding:** This study was supported by Bozhou City Key R&D Program of Anhui Province of China (No. bzzc2024004 & bzzc2024001), Henan Provincial Science and Technology Research Projects (Grant No. 212102310799 and 222102310577). MR Wang contributed to the conception and design of the study, data collection, and manuscript writing. TS provided expertise in statistical analysis, assisted in drafting the statistical methodology section and supervised the entire research project. RY Jiang involved in data analysis, interpretation of results, and critical revision of the manuscript. FD Hu Assisted with data collection and laboratory analysis.TS and MR Wang confirm the authenticity of all the raw data.

**Competing interests:** The authors have declared that no competing interests exist.

## Conclusions

The Lp(a)-CatLet© synergy enhances MACCE risk stratification in ePCI-treated AMI, reflecting complementary *pathobiological* (Lp(a)-driven plaque vulnerability) and *anatomical* (CatLet©-quantified complexity) pathways. This dual-parameter approach could support post-PCI risk stratification and follow-up planning.

## 1. Introduction

Accurate risk stratification following emergency revascularization for AMI remains a critical challenge in contemporary cardiology practice. While current prognostic tools incorporating biochemical markers (e.g., cardiac troponins) and angiographic indices (e.g., SYNTAX score) provide modest predictive value [1,2], their individual limitations—including biological variability and anatomical oversimplification—underscore the need for more robust multimodal approaches [3,4].

Emerging evidence implicates lipoprotein(a) (Lp(a)), a proatherogenic and prothrombotic particle, as a key mediator of post-PCI complications [5]. Concurrently, the CatLet© scoring system (www.catletscore.com) has demonstrated superior discriminatory capacity for coronary lesion complexity compared to conventional scoring systems by incorporating dynamic weighting for bifurcation lesions, calcification, and functional significance [6].

We hypothesized that integrating Lp(a)'s pathobiological insights with CatLet©'s anatomical precision would optimize MACCE prediction. This investigation represents the first evaluation of this dual-parameter model in ePCI-AMI patients.

## 2. Materials and methods

### 2.1 Study design and population

This retrospective cohort study included 307 consecutive acute myocardial infarction (AMI) patients undergoing successful emergency percutaneous coronary intervention (ePCI) at Bozhou People's Hospital (China) between January 2020 and May 2022 (Fig 1). AMI was diagnosed per the Fourth Universal Definition of Myocardial Infarction. Key exclusion criteria: prior MI, chronic heart failure (LVEF <40%), severe renal dysfunction (eGFR < 30 mL/min/1.73m²), or non-atherosclerotic coronary disease (e.g., vasculitis, cardiomyopathy). The study was conducted in accordance with the Declaration of Helsinki. The protocol approval was obtained from the Medical Ethics Committee of Bozhou Hospital affiliated with Anhui Medical University(No. bz-2020–004). The identity of patients remained anonymous, and the requirement for informed consent was waived due to the retrospective anonymized data analysis.

### 2.2 Procedural and pharmacological protocols

All patients received guideline-directed therapy:

Pre-PCI: Loading doses of aspirin (300 mg) plus P2Y12 inhibitor (clopidogrel 300/600 mg or ticagrelor 180 mg).

## Figure 1 Study Design & Population Selection
### Design: Retrospective cohort study

**Inclusion Criteria:**
AMI patients undergoing successful ePCI (n=307)
Adherence to *2021 Chinese Expert Consensus on AMI*

**Exclusion Criteria:**
Age >90 years, prior MI, chronic heart failure, severe renal/hepatic dysfunction, or other confounding conditions.

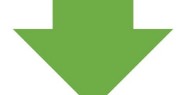

## Data Collection & Group Stratification

**Baseline Data:**
Demographics (age, gender, comorbidities)
Laboratory tests: LP(a), cTnI, NT-proBNP (fasting blood samples post-admission)
Angiographic scores: CatLet© and Gensini (via www.catletscore.com)

**Group Stratification:**
**Observation Group (n=78):** Patients with 1-year MACCE (cardiac death, recurrent MI, TLR, stroke)
**Control Group (n=229):** MACCE-free patients

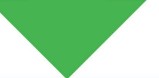

## Laboratory & Angiographic Analysis

**LP(a) Measurement:**
Immunoturbidimetric assay (Fuzhou Lvchuan Biotechnology, China)

**Angiographic Scoring:**
**CatLet© Score:** Dynamic weighting for bifurcation/calcification/functional significance.
**Gensini Score:** Traditional stenosis severity assessment.

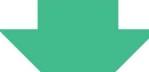

## Statistical Analysis:
### Primary Analysis:

**Univariate/Multivariate Logistic Regression:** Identified predictors of MACCE (LP(a), CatLet© score, LVEF, cTnI, etc.).

**ROC Curves:** Evaluated predictive performance (AUC, sensitivity, specificity) for LP(a), CatLet©, and combined model.

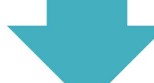

## Secondary Analysis:

Net reclassification improvement (NRI) and bootstrap validation.

**Fig 1. Study design and population selection.**

Anticoagulation: Intravenous unfractionated heparin (100 IU/kg; ACT > 300 sec), with glycoprotein IIb/IIIa inhibitors at operator discretion.

Post-PCI: Dual antiplatelet therapy (aspirin 100 mg/day + clopidogrel 75 mg/day or ticagrelor 90 mg twice daily) for ≥12 months, with high-intensity statins and beta-blockers as indicated.

High-risk support: IVUS-guided PCI for complex lesions (e.g., severe calcification, bifurcation).

## 2.3 Biomarker and angiographic analysis

Lp(a) quantification: Fasting serum Lp(a) was measured 24h post-admission via immunoturbidimetric assay (Roche; inter-assay CV < 5%) on a Roche (Cobas c 501 analyzer).(On the morning of the second day after admission, venous blood was drawn from the patient [7]. Blood sample preparation involved drawing approximately 4 mL of venous blood from the median cubital vein under sterile conditions, after the patient had fasted overnight. The sample was allowed to stand for 20 minutes before being centrifuged at 4000 rpm to separate the serum. The serum was then stored at −80°C for subsequent measurement of Lp(a) concentration.)

## 2.4 Angiographic scoring

**Gensini score.** Lesion severity was graded based on coronary stenosis degree, with segment-specific weighting factors applied. The total score was derived from the sum of individual lesion scores [8].

**CatLet© coronary score.** Angiographic complexity was assessed using the web-based CatLet© scoring calculator (available at http://www.catletscore.com). Only stenotic lesions were quantified [6,9].

Two independent interventional cardiologists, blinded to all clinical and laboratory data, performed the CatLet© Coronary and Gensini scoring. Both analysts completed a standardized training module provided by the CatLet© Coronary Score developers. Inter-observer reliability was excellent, with an intraclass correlation coefficient (ICC) of 0.92 (95% CI: 0.88–0.95) for the continuous CatLet© Coronary Score and a Cohen's κ of 0.85 for categorical lesion feature assessment (e.g., bifurcation type, severe calcification). Discrepancies were resolved by a third senior analyst.

**Follow-up.** Follow-up was conducted via clinical visits and telephone interviews at 1, 6, and 12 months. Loss to follow-up was < 5%. The censoring date was May 31, 2023, with yearly assessments continuing thereafter. A routine angiographic evaluation was advised at the 12-month mark following PCI. If patients experienced a return of angina symptoms, angiography was conducted earlier. For those who did not follow the recommended follow-up schedule, telephone interviews were conducted [10].

## 2.5 Endpoint definitions [10,11]

The main outcome of this research was a patient-focused composite measure of major adverse cardiac and cerebrovascular events (MACCE) at 1-year, encompassing all-cause mortality, myocardial infarction (MI), and revascularization of the target vessel or lesion (TVR/TLR). When evaluating cumulative outcomes, each event was counted only once, prioritizing the first occurrence.

The secondary outcomes included a safety composite endpoint of all-cause death, MI, and stroke, alongside individual MACCE components and stent thrombosis (ST).

Death was categorized as any post-procedure fatality, classified into cardiac or non-cardiac based on the Academic Research Consortium (ARC) guidelines. A death was presumed to be of cardiac origin unless proven otherwise. Cardiac death encompassed fatalities due to cardiac issues (e.g., MI, low-output heart failure, lethal arrhythmias), those related to the procedure, or those of unknown origin.

Myocardial infarction (MI) was defined in accordance with the fourth universal MI definition [12], emphasizing elevated cardiac troponin (cTn) levels exceeding the 99th percentile upper reference limit (URL), symptoms of myocardial ischemia, ECG changes, and angiographic findings. Within 48 hours post-procedure, cTn levels exceeding 5 times the 99th

percentile URL after PCI or 10 times after CABG indicated periprocedural MI in patients with normal baseline cTn levels. If baseline levels were elevated, stable, or decreasing, a cTn increase of over 20% also signified periprocedural PCI-related MI. Q-wave MI was identified by the presence of a new pathological Q-wave in at least two contiguous leads after initial treatment.

TVR was any surgical or percutaneous revascularization of any segment within the stented vessel (including target lesions and adjacent branches) within a year, such as the left main, left anterior descending, and left circumflex arteries. A pre-planned staged PCI was not deemed a TVR. TVR was clinically driven, with routine angiographic follow-up not counted as an event unless accompanied by ischemic symptoms or objective evidence of ischemia.

Definite stent thrombosis (ST) was defined per ARC criteria for PCI, while graft occlusion followed an ARC-like definition for CABG.

The definition of stroke is based on the currently internationally recognized criteria primarily derived from the expert consensus statement of the American Heart Association/American Stroke Association (AHA/ASA) [13].

**Core definition.** Stroke is defined as central nervous system (CNS) cell death accompanied by permanent injury, caused by ischemia or hemorrhage, based on neuropathological, neuroimaging, and/or clinical evidence.

**Scope of coverage.** This includes ischemic stroke (symptomatic CNS infarction), asymptomatic cerebral infarction (CNS infarction without known clinical symptoms), intracerebral hemorrhage, and subarachnoid hemorrhage.

### 2.6 Statistical analysis

Continuous variables (mean±SD or median[IQR]) were compared via Student's *t*-test or Mann-Whitney U test; categorical variables (n[%]) via χ² or Fisher's exact test. Multivariable logistic regression (adjusted for age, sex, hypertension, diabetes, and LVEF) identified MACCE predictors. Multicollinearity was assessed using Variance Inflation Factors (VIF); all VIF values were <2.5, indicating no significant collinearity.

Separate models with CatLet and Gensini scores were compared using Akaike Information Criterion (AIC).ROC analysis determined optimal Lp(a)/CatLet© cutoffs. Statistical significance was set at two-tailed P<0.05. Analyses used SPSS v26.0 (IBM).

## 3. Results

### 3.1 Baseline characteristics

The MACCE group exhibited lower LVEF (51.6±11.7% vs. 54.9±12.0%, P=0.038), higher peak cTnI (4.86±2.13 vs. 4.35±1.56 ng/mL, P=0.023), and elevated Lp(a) (135.99±33.07 vs. 123.35±42.70 nmol/L, P=0.018) (Table 1).

### 3.2 Angiographic characteristics were comparable except for higher CatLet© (33.6±9.0 vs. 30.8±8.2, P=0.001) and Gensini scores (83.9±16.9 vs. 78.5±20.7, P=0.042) in the MACCE group (Table 2)

Multivariable logistic regression confirmed Lp(a) (OR = 2.339, P<0.001) and CatLet© score (OR = 1.092, P=0.005) as independent predictors of MACCE, along with LVEF, cTnI, and Gensini score (Table 3). Consistent results were observed in Cox proportional-hazards analysis (Table 4). Log-transformed Lp(a) as a continuous variable also showed a significant dose-response relationship with MACCE risk.All Variance Inflation Factors (VIF) were<2.5, indicating no concerning multicollinearity (Table 5).

Assignment: LVEF (≥50%=0,+1 for every 5% decrease); cTnI (0 points,+0.5 for every 1 ng/mL increase); Lp(a) (≥70.7 nmol/L=1,<70.7 nmol/L=0); Gensini Score (0~10 points=0,+1 for every 10-point increase); CatLet© Coronary Score (0~5 points=0,+1 for every 5-point increase, additional +1 for presence of calcification/angulation).

The model was adjusted for age, sex, hypertension, and diabetes mellitus. Collinearity was not a concern (all Variance Inflation Factors<2.5). The model demonstrated good calibration (Hosmer-Lemeshow goodness-of-fit test, χ²=8.12, P=0.422).

**Table 1. Baseline characteristics of the two groups [n (%), (`x±s)].**

| | Case Group (n=78) | Control Group (n=229) | t/χ2 | P-value |
|---|---|---|---|---|
| Age (years) | 66.17±9.85 | 63.41±12.34 | 1.9707 | 0.0743 |
| Gender(Male/Female) | 55/23 | 165/64 | 0.0679 | 0.7944 |
| Hypertension | 31 (39.7) | 100 (43.7) | 0.3663 | 0.545 |
| Systolic Blood Pressure (mmHg) | 135.38±27.28 | 129.79±25.47 | 1.6462 | 0.1008 |
| Diastolic Blood Pressure (mmHg) | 76.36±16.58 | 80.41±17.04 | 1.8241 | 0.0691 |
| Diabetes# | 10 (12.8) | 28 (12.2) | 0.0189 | 0.8907 |
| LDL-C(mmol/L) | 2.78±1.17 | 2.69±1.28 | 0.5487 | 0.5836 |
| Smoking(%) | 18 (23.1) | 56 (24.5) | 0.0603 | 0.806 |
| Alcohol Consumption(%) | 11 (14.1) | 27 (11.8) | 0.2868 | 0.5923 |
| Peripheral Arterial Disease | 2 (2.6) | 5 (2.2) | 0.0598 | 0.8068 |
| HBA1c | 4.72±1.17 | 4.69±1.26 | 0.1272 | 0.8989 |
| Serum Creatinine (μmol/L) | 73.17±18.68 | 75.26±22.37 | 0.7418 | 0.4588 |
| Killip class (≥2) | 22 (28.2) | 45 (19.7) | 2.4956 | 0.1142 |
| LVEF(%) | 51.64±11.71 | 54.89±11.95 | 2.0854 | 0.0379* |
| cTnI (ng/ml) | 4.86±2.13 | 4.35±1.56 | 2.2933 | 0.0225* |
| NT-proBNP (pg/mL) | 857.64±274.37 | 791.18±255.64 | 1.946 | 0.0526 |
| LP(a) (nmol/L) | 135.99±33.07 | 123.35±42.70 | 2.3826 | 0.0178* |
| Medication[n (%)] | | | | |
| Clopidogrel | 60 (76.9) | 155 (67.7) | 2.3656 | 0.124 |
| Ticagrelor | 18 (23.1) | 74 (32.3) | 2.3656 | 0.124 |
| ACEI/ARB | 50 (64.1) | 160 (69.9) | 0.8951 | 0.3441 |
| Tirofiban | 26 (33.3) | 58 (25.3) | 1.8763 | 0.1708 |
| β-Blocker | 43 (55.1) | 140 (61.1) | 0.872 | 0.3504 |
| Pro-urokinase | 8 (11.5) | 25 (10.9) | 0.0228 | 0.88 |

Note: Comparisons between the low group and either the medium or high group yielded a P<0.05, while comparisons between the medium and high groups revealed b P<0.05. LDL-C refers to Low-Density Lipoprotein Cholesterol; HBA1c, Glycated Hemoglobin; LVEF, Left Ventricular Ejection Fraction; and NT-proBNP, N-Terminal Pro-B-type Natriuretic Peptide; ACEI/ARB: angiotensin-converting enzyme inhibitors/angiotensin receptor blockers.

### 3.3 Predictive Performance of Lp(a) and CatLet© Score for MACCE

In this cohort of AMI patients undergoing emergency PCI, both Lp(a) and the CatLet© coronary angiographic score demonstrated significant prognostic utility for MACCE: Individual Predictors: Lp(a):AUC: 0.780 (95% CI: 0.745–0.896),Optimal cutoff: 70.70 nmol/L (sensitivity 81.0%, specificity 76.5%, P<0.001),Clinical implication: Exceeds the EAS consensus high-risk threshold (125 nmol/L)(7), suggesting potential utility in risk stratification. CatLet© Score:AUC: 0.833 (95% CI: 0.820–0.927),Optimal cutoff: 18.6 points (sensitivity 83.0%, specificity 79.0%, P<0.001),Anatomic correlation: Aligns with prior validation studies defining scores ≥18 as indicative of complex coronary anatomy. Combined Model Superiority: The integration of Lp(a) and CatLet© score yielded enhanced predictive performance: AUC: 0.862 (95% CI: 0.830–0.960),Sensitivity/specificity: 88.0%/83.0% (P=0.009 vs. individual predictors, DeLong's test confirmed the superiority of the combined model over individual predictors (P = 0.0089, Z = 2.64 vs. Lp(a); P = 0.034, Z = 2.12 vs. CatLet© score).Net reclassification improvement (NRI): 0.35 (95% CI: 0.12–0.58; P=0.002),Statistical robustness: Bootstrap validation (1,000 iterations) confirmed model stability (mean AUC ± SD: 0.852±0.021) (Table 6) (Fig 2). The combined model showed excellent calibration across risk deciles (Hosmer-Lemeshow P = 0.422; Table 7).

**Table 2. Angiographic and procedural characteristics of ePCI-AMI subgroups.**

| Variable | Case Group (n = 78) | Control Group (n = 229) | t/x | P值 |
|---|---|---|---|---|
| Treated vessel | | | | |
| LM/LAD/D1, n (%) | 35(44.9) | 94(41.0) | 0.3492 | 0.5546 |
| LCX/OM, n (%) | 18(23.0) | 71(31.1) | 1.7762 | 0.1826 |
| RCA #, n (%) | 25(32.1) | 64(27.9) | 0.476 | 0.4902 |
| Triple-vessel disease | 8(10.1) | 20(8.7) | 0.1628 | 0.6866 |
| Lesion characteristics and coronary anatomy | | | | |
| Bifurcation Medina classification, n (%) | 56 (71.8) | 154 (67.2) | 0.5563 | 0.4558 |
| Medina 1, 1, 1 | 5 (8.9) | 14 (9.1) | 0.0013 | 0.9712 |
| Medina 1, 1, 0 | 6 (10.7) | 15 (9.7) | 0.0433 | 0.8352 |
| Medina 1, 0, 1 | 5 (6.4) | 9 (5.8) | 0.23 | 0.6315 |
| Medina 1, 0, 0 | 4 (5.1) | 6 (3.9) | 0.3084 | 0.5787 |
| Medina 0, 1,1 | 16 (20.5) | 28 (18.2) | 2.6765 | 0.1018 |
| Medina 0, 1, 0 | 11 (19.6) | 44 (28.6) | 1.6935 | 0.1931 |
| Medina 0, 0, 1 | 9 (16.1) | 38 (24.7) | 1.75 | 0.1859 |
| Severe calcification | 5 (6.4) | 20 (8.7) | 0.4199 | 0.517 |
| Tortuosity | 8 (12.8) | 25 (10.9) | 0.0265 | 0.8707 |
| Thrombus | 6 (7.7) | 14 (6.1) | 0.2381 | 0.6256 |
| LAD subtypes: | | | | |
| Short LAD | 11 (31.4) | 33 (35.1) | 0.1535 | 0.6952 |
| Long LAD | 14 (40.0) | 32 (34.0) | 0.3945 | 0.5299 |
| Normal LAD | 10 (28.6) | 29 (30.8) | 0.0628 | 0.8021 |
| Diagonal branch size: | | | | |
| Small diagonal (≤2.0 mm) | 5 (14.3) | 14 (14.9) | 0.005 | 0.9436 |
| Medium diagonal (2.0–2.5 mm) | 10 (28.5) | 40 (42.6) | 0.5919 | 0.4417 |
| Large diagonal (>2.5 mm) | 6 (17.1) | 16 (17.0) | 0.2877 | 0.5917 |
| RCA dominance D: | | | | |
| Non-PDA type | 1 (4.0) | 2 (3.1) | 0.2044 | 0.8381 |
| Single PDA supply | 2 (8.0) | 4 (6.3) | 0.4012 | 0.6883 |
| Small RCA (≤2.0 mm) | 8 (32.0) | 18 (28.1) | 0.1305 | 0.7179 |
| Medium RCA (2.0–2.9 mm) | 11 (44.0) | 30 (46.7) | 0.0598 | 0.8068 |
| Large RCA (3.0–4.0 mm) | 2 (8.0) | 6 (9.4) | 0.2027 | 0.8394 |
| Very large RCA (>4.0 mm) | 1 (4.0) | 4 (6.3) | 0.412 | 0.6804 |
| Lesion length >20 mm | 24 (30.1) | 74 (32.3) | 0.0639 | 0.8004 |
| Stents implanted | | | | |
| Number | 1.81 ± 0.56 | 1.69 ± 0.49 | 1.7076 | 0.0897 |
| Max stent Ø (mm) | 2.80 ± 0.36 | 2.83 ± 0.42 | 0.6314 | 0.5283 |
| cumulative stent length, mm | 28.62 ± 7.68 | 26.62 ± 7.98 | 1.9214 | 0.0556 |
| Procedure duration (min) | 46.32 ± 13.44 | 43.53 ± 10.90 | 1.8371 | 0.0672 |
| Total contrast volume (ml) | 84.06 ± 15.35 | 80.12 ± 19.17 | 1.6444 | 0.1011 |
| Preprocedural TIMI flow grade ≤2# | 60 (76.9) | 167 (72.9) | 0.4825 | 0.4873 |
| Ischemia time (h) | 2.7 ± 0.7 | 2.6 ± 0.9 | 0.998 | 0.3191 |
| MBG | 2.86 ± 0.39 | 2.92 ± 0.29 | 1.3905 | ? |
| Gensini score | 83.85 ± 16.93 | 78.54 ± 20.68 | 2.045 | 0.0417 |
| CatLet© coronary score | 33.58 ± 9.04 | 30.80 ± 8.24 | 3.34 | 0.0012 |

Note: Statistical analysis: count data are presented as n (%), and measurement data as mean ± standard deviation (SD) unless otherwise specified. Group comparisons were performed using Fisher's exact test (for categorical data) or Pearson's χ² test (for continuous data). Two-tailed testing P value

*(Continued)*

**Table 2.** (Continued)

<0.05 was considered statistically significant. LM/LAD/D1: Left Main/Left Anterior Descending/First Diagonal Branch.OM1/OM2: First Obtuse Marginal/ Second Obtuse Marginal.RCA:Right Coronary Artery.Δ: According to the 2023 European Society of Cardiology guidelines for the management of acute coronary syndromes: Statement of endorsement by the NVVC. Neth Heart J. 2024 Oct;32(10):338–345.:TIMI trial (NEJM 1985;312:932–936). MB-G:;Myocardial Blush Grade (Eur Heart J 2022;43:378–389).

**Table 3. Multivariable logistic regression analysis of factors influencing MACCE.**

| Variable | β (Coefficient) | SE (Standard Error) | Wald Value | P Value | OR (Odds Ratio) | 95% CI for OR |
|---|---|---|---|---|---|---|
| LVEF (per 1-score decrease) | −0.43 | 0.11 | 15.29 | <0.001 | 0.65 | 0.524 - 0.806 |
| cTnI (per 0.5-score increase) | 0.23 | 0.052 | 19.55 | <0.001 | 1.259 | 1.136 - 1.395 |
| Lp(a) (≥70.7 vs.<70.7 nmol/L) | 0.85 | 0.22 | 14.91 | <0.001 | 2.339 | 1.519 - 3.603 |
| Gensini Score (per 10-point) | 0.041 | 0.01 | 16.81 | <0.001 | 1.042 | 1.021 - 1.063 |
| CatLet© Score (per 1-score) | 0.088 | 0.031 | 8.05 | 0.005 | 1.092 | 1.027 - 1.161 |

Note: MACCE = Major Adverse Cardiac and Cerebrovascular Events; LVEF = Left Ventricular Ejection Fraction; cTnI = Cardiac Troponin I; Lp(a) = Lipoprotein(a); OR = Odds Ratio; CI = Confidence Interval.

**Table 4. Cox Proportional-Hazards Regression Analysis for 1-Year MACCE.**

| Variable | β (Coefficient) | SE | Wald | P-value | HR (Hazard Ratio) | 95% CI for HR |
|---|---|---|---|---|---|---|
| LVEF (per 5% decrease) | −0.421 | 0.108 | 15.18 | <0.001 | 0.656 | 0.531–0.811 |
| cTnI (per 1 ng/mL increase) | 0.225 | 0.051 | 19.42 | <0.001 | 1.252 | 1.133–1.384 |
| Lp(a) (≥70.7 vs.<70.7 nmol/L) | 0.832 | 0.215 | 14.98 | <0.001 | 2.298 | 1.507–3.505 |
| Gensini Score (per 10-point) | 0.04 | 0.01 | 16 | <0.001 | 1.041 | 1.021–1.062 |
| CatLet© Score (per 1-point) | 0.085 | 0.03 | 8.02 | 0.005 | 1.089 | 1.027–1.155 |

Notes:MACCE: Major Adverse Cardiac and Cerebrovascular Events. LVEF: Left Ventricular Ejection Fraction. cTnI: Cardiac Troponin I. Lp(a): Lipoprotein(a). HR: Hazard Ratio. CI: Confidence Interval. Model adjusted for age, sex, hypertension, and diabetes mellitus. Proportional hazards assumption was satisfied (Schoenfeld residual test, P > 0.05 for all covariates).The model demonstrates consistent predictive performance with the logistic regression model, supporting the robustness of Lp(a) and CatLet© score as independent predictors of time-to-MACCE.

**Table 5. Multivariable logistic regression analysis with log-transformed Lp(a) as a continuous predictor for MACCE.**

| Variable | β (Coefficient) | SE | Wald | P-value | OR (Odds Ratio) | 95% CI for OR |
|---|---|---|---|---|---|---|
| Log-Lp(a) (per 1-SD increase) | 0.671 | 0.181 | 13.74 | <0.001 | 1.956 | 1.373–2.787 |
| LVEF (per 5% decrease) | −0.425 | 0.109 | 15.22 | <0.001 | 0.654 | 0.528–0.810 |
| cTnI (per 1 ng/mL increase) | 0.227 | 0.051 | 19.78 | <0.001 | 1.255 | 1.136–1.387 |
| Gensini Score (per 10-point) | 0.039 | 0.01 | 15.21 | <0.001 | 1.04 | 1.020–1.060 |
| CatLet© Score (per 1-point) | 0.086 | 0.03 | 8.21 | 0.004 | 1.09 | 1.028–1.156 |

Notes:Lp(a) was natural log (ln)-transformed to approximate a normal distribution due to its skewed nature. The OR represents the change in odds of MACCE associated with a 1-standard deviation (SD) increase in ln-Lp(a).The SD of ln-Lp(a) in this cohort was 0.89.The model was adjusted for age, sex, hypertension, and diabetes mellitus.All Variance Inflation Factors (VIF) were < 2.5, indicating no concerning multicollinearity.This analysis confirms a significant, dose-response relationship between continuously scaled Lp(a) levels and the risk of MACCE, which strengthens the primary findings and mitigates concerns regarding arbitrary dichotomization.

**Table 6. Analysis of the predictive efficacy of Lp(a) levels and CatLet scores for MACCE in AMI patients.**

| Parameter | AUC | Youden Index | Cutoff Value | Sensitivity (%) | Specificity (%) | P-value | 95%CI |
|---|---|---|---|---|---|---|---|
| Lp(a) [a] | 0.78 | 0.575 | 70.7 | 81 | 76.5 | <0.001 | 0.745-0.896 |
| CatLet© coronary score | 0.833 | 0.62 | 18.6 | 83 | 79 | <0.001 | 0.820-0.927 |
| Combined model[b] | 0.862 | 0.71 | | 88 | 83 | 0.0089 | 0.83-0.96 |

Footnotes:a) Lp(a) measured by immunoturbidimetric assay (Roche Diagnostics, intra-assay CV < 5%).b) The combined model (Lp(a) + CatLet© score) demonstrated superior predictive efficacy compared to individual parameters. DeLong's test confirmed the superiority of the combined model over individual predictors (P = 0.0089, Z = 2.64 vs. Lp(a); P = 0.034, Z = 2.12 vs. CatLet© score).

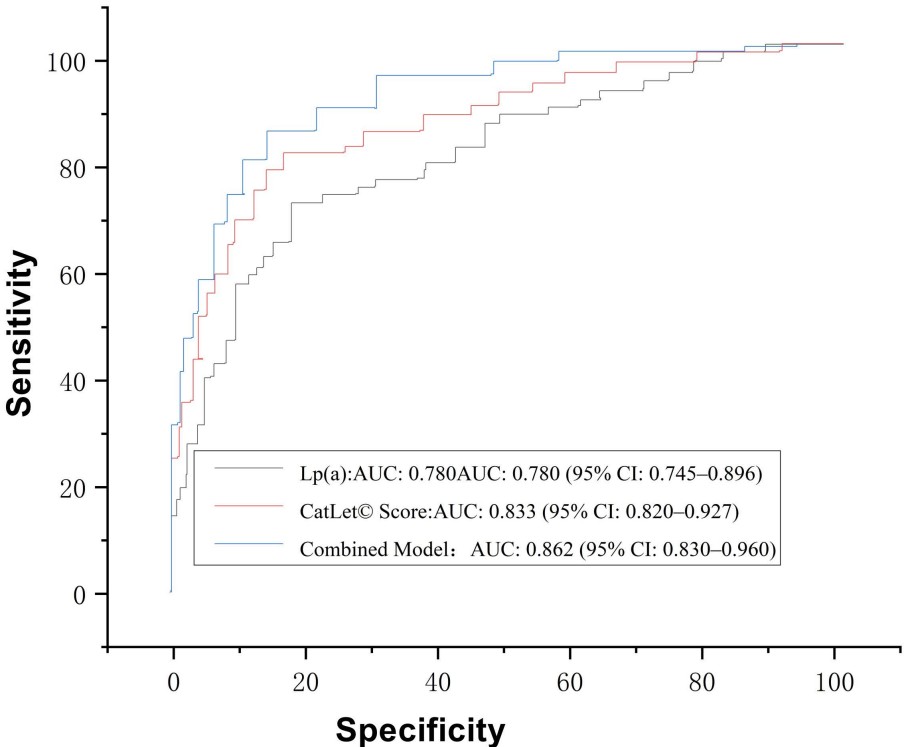

**Fig 2. ROC analysis of Lp(a) and CatLet score for 1-Year MACCE prediction post-AMI.**

## 4. Discussion

This study demonstrates that the integration of lipoprotein(a) [Lp(a)] and the CatLet© coronary angiographic score significantly enhances the prediction of 1-year MACCE in AMI patients undergoing emergency PCI. The combined model outperformed either parameter alone, achieving an AUC of 0.862, with improved sensitivity and specificity. These findings underscore the complementary value of incorporating both a pathobiologic marker of plaque vulnerability and a comprehensive anatomic-functional scoring system into post-PCI risk stratification.

### 4.1. Synergistic predictive value of Lp(a) and CatLet© Score

The independent association of elevated Lp(a) (≥70.70 nmol/L) with MACCE aligns with its well-established proatherogenic, prothrombotic, and proinflammatory properties [5,14]. Lp(a) promotes plaque instability through mechanisms

**Table 7. Calibration of the combined logistic regression model (Lp(a) + CatLet© Score) for Predicting 1-Year MACCE.**

| Decile of Predicted Risk | Number of Patients | Number of Events (Observed) | Observed Event Rate (%) | Mean Predicted Risk (%) |
|---|---|---|---|---|
| 1 (Lowest Risk) | 31 | 2 | 6.5 | 5.2 |
| 2 | 31 | 3 | 9.7 | 8.9 |
| 3 | 31 | 4 | 12.9 | 12.5 |
| 4 | 30 | 5 | 16.7 | 16.3 |
| 5 | 31 | 6 | 19.4 | 20.1 |
| 6 | 31 | 7 | 22.6 | 24.2 |
| 7 | 31 | 9 | 29 | 29 |
| 8 | 30 | 11 | 36.7 | 34.8 |
| 9 | 31 | 14 | 45.2 | 42.5 |
| 10 (Highest Risk) | 30 | 17 | 56.7 | 55.1 |

Notes:The combined model includes Lp(a) (≥70.7 nmol/L), CatLet© Score, LVEF, cTnI, and Gensini Score, adjusted for age, sex, hypertension, and diabetes.Calibration assessment: The close agreement between the Observed Event Rate and the Mean Predicted Risk across all deciles indicates excellent calibration of the model. Hosmer-Lemeshow goodness-of-fit test: $\chi^2 = 8.12$, $P = 0.422$. A non-significant P-value indicates no significant deviation between the predicted and observed risks, confirming good model calibration.

including the inhibition of fibrinolysis, enhancement of foam cell formation, and induction of endothelial dysfunction. Concurrently, the CatLet© score provided a granular assessment of coronary lesion complexity, incorporating critical features such as bifurcation anatomy, calcification severity, and functional significance—elements not fully captured by traditional scores like SYNTAX or Gensini [6,9,15].

The superior predictive performance of the combined model suggests a synergistic interaction between systemic risk (reflected by Lp(a)) and localized anatomic burden (quantified by CatLet©). Patients with high Lp(a) may exhibit a heightened inflammatory and thrombotic milieu, which, when coupled with complex coronary anatomy, predisposes them to recurrent ischemic events. This dual-parameter approach thus bridges a critical gap between circulating risk biomarkers and coronary morphology.

## 4.2. Comparative advantages of the CatLet© score

Our results reinforce the clinical utility of the CatLet© score over conventional angiographic scoring systems. While the Gensini score focuses primarily on stenosis severity, and the SYNTAX score emphasizes lesion complexity, the CatLet© system incorporates dynamic weighting for calcification, angulation, and functional parameters such as TIMI flow—factors directly relevant to PCI difficulty and prognosis [6,16]. While the SYNTAX score remains a widely adopted tool, CatLet integrates functional and morphological features that may enhance risk stratification in AMI. And,its open-access, web-based platform (www.catletscore.com) and the high inter-observer reliability demonstrated in this and prior studies facilitate its implementation in clinical practice with appropriate training.As shown in Table 8 and 9, the CatLet© score offers a more holistic evaluation, which may explain its stronger discriminatory capacity in our cohort.

## 4.3. Clinical and mechanistic implications

From a pathophysiological perspective, Lp(a) has been implicated in vascular calcification and fibrous cap thinning [17–19], processes that may exacerbate the anatomic complexity quantified by the CatLet© score. This interplay suggests that patients with elevated Lp(a) and high CatLet© scores may represent a high-risk phenotype warranting intensified management. This could include more aggressive lipid-lowering therapy (e.g., PCSK9 inhibitors), prolonged dual antiplatelet therapy, or closer follow-up—though such strategies require validation in prospective trials.

 

**Table 8. Comparative advantages of CatLet© versus SYNTAX/Gensini scoring systems.**

| Scoring System | Key Focus | Limitations | Improvements in CatLet© |
|---|---|---|---|
| SYNTAX | Lesion complexity & quantity | Lacks functional assessment; cumbersome to use | Integrates FFR/TIMI flow; streamlined workflow |
| Gensini | Stenosis severity & plaque burden | Omits calcification, thrombus assessment | Incorporates morphology (calcification, angulation) |
| CatLet© | Anatomic + functional + PCI difficulty criteria | Requires multicenter validation (ongoing) | Dynamic weight adjustment for real-world PCI decision-making* |

Footnotes: FFR: Fractional Flow Reserve; TIMI: Thrombolysis in Myocardial Infarction flow grade Dynamic weights adjust for lesion location (e.g., left main), morphology (calcification/angulation), and functional significance.

**Table 9. Comparative diagnostic performance of CatLet©, SYNTAX, and Gensini Scores for Predicting 1-Year MACCE.**

| Scoring System | AUC (95% CI) | Optimal Cut-off | Sensitivity (%) | Specificity (%) | P-value | DeLong's Test P-value (vs. CatLet©) |
|---|---|---|---|---|---|---|
| CatLet© Score | 0.833 (0.820–0.927) | ≥ 18.6 | 83 | 79 | < 0.001 | (Reference) |
| SYNTAX Score | 0.798 (0.765–0.901) | ≥ 23 | 78.2 | 75.1 | < 0.001 | 0.045 |
| Gensini Score | 0.782 (0.750–0.885) | ≥ 72 | 76.9 | 73.8 | < 0.001 | 0.012 |
| Combined Model (Lp(a) + CatLet©) | 0.862 (0.830–0.960) | – | 88 | 83 | < 0.001 | – |

Note:AUC: Area Under the Receiver Operating Characteristic Curve.The SYNTAX score was retrospectively calculated for all 307 patients by the same two blinded analysts using standard methodology.The DeLong's test was used to compare the AUC of each traditional score against the CatLet© score. The combined model refers to the integration of Lp(a) (cut-off ≥70.70 nmol/L) and the CatLet© score (cut-off ≥18.6) in a logistic regression model.

Moreover, the optimal Lp(a) cutoff of 70.70 nmol/L in our cohort is notably lower than the European Atherosclerosis Society consensus threshold of 125 nmol/L [7], and should be considered hypothesis-generating, requiring validation in multicenter cohorts.This may reflect differences in study populations or the acute clinical context, highlighting the need for context-specific thresholds in AMI settings.

### 4.4. Limitations

Several limitations should be acknowledged. First, the single-center, retrospective design introduces potential selection bias. Second, although the CatLet© score has been validated in prior studies [6], its application requires specialized training and is not yet widely adopted. Third, Lp(a) assays are not fully standardized across platforms, which may affect the generalization of the proposed cutoff [7,19]. Finally, unmeasured confounders such as genetic predispositions or residual inflammatory risk may influence outcomes.

### 4.5. Future directions

Prospective multicenter studies are needed to validate our findings and refine risk thresholds. Incorporating emerging biomarkers (e.g., high-sensitivity C-reactive protein) or artificial intelligence–based plaque characterization could further enhance predictive models [20]. Additionally, randomized trials are warranted to assess whether risk-guided therapeutic interventions improve outcomes in this high-risk subgroup.

### 4.6. Conclusion

In conclusion, the combination of Lp(a) and the CatLet© score provides a robust, clinically feasible tool for stratifying MACCE risk in AMI patients after emergency PCI. This integrated model captures both systemic lipid-related risk and

coronary anatomic complexity, offering a more comprehensive prognostic assessment than either measure alone. With further validation, this approach could inform personalized secondary prevention strategies and follow-up protocols.

## Supporting information

**S1 File. Graphical abstract.**
(TIFF)

## Author contributions

**Conceptualization:** Mengru Wang, Sheng tu.

**Data curation:** Mengru Wang, Fudong Hu, Rongyan Jiang.

**Formal analysis:** Mengru Wang, Rongyan Jiang.

**Funding acquisition:** Sheng tu, Rongyan Jiang.

**Investigation:** Mengru Wang, Sheng tu.

**Methodology:** Mengru Wang, Sheng tu, Fudong Hu.

**Project administration:** Sheng tu.

**Resources:** Mengru Wang, Sheng tu.

**Software:** Fudong Hu, Rongyan Jiang.

**Supervision:** Mengru Wang, Rongyan Jiang.

**Validation:** Mengru Wang, Sheng tu, Fudong Hu.

**Visualization:** Mengru Wang, Sheng tu.

**Writing – original draft:** Sheng tu.

**Writing – review & editing:** Mengru Wang.

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
