## [Decision Letter · Decision Letter 0]

18 Dec 2025

Dear Dr. tu,

Thank you for submitting your manuscript to PLOS ONE. After careful consideration, we feel that it has merit but does not fully meet PLOS ONE’s publication criteria as it currently stands. Therefore, we invite you to submit a revised version of the manuscript that addresses the points raised during the review process.

We look forward to receiving your revised manuscript.

Kind regards,

Timir Paul

Academic Editor

PLOS One

**Journal Requirements:**

1. When submitting your revision, we need you to address these additional requirements. Please ensure that your manuscript meets PLOS ONE's style requirements, including those for file naming. The PLOS ONE style templates can be found at https://journals.plos.org/plosone/s/file?id=wjVg/PLOSOne_formatting_sample_main_body.pdf and https://journals.plos.org/plosone/s/file?id=ba62/PLOSOne_formatting_sample_title_authors_affiliations.pdf 2. Please update your submission to use the PLOS LaTeX template. The template and more information on our requirements for LaTeX submissions can be found at http://journals.plos.org/plosone/s/latex. 3. We note that the grant information you provided in the ‘Funding Information’ and ‘Financial Disclosure’ sections do not match.  When you resubmit, please ensure that you provide the correct grant numbers for the awards you received for your study in the ‘Funding Information’ section. 4. Please upload a new copy of Figure 1 as the detail is not clear. Please follow the link for more information:  https://journals.plos.org/plosone/s/figures 5. If the reviewer comments include a recommendation to cite specific previously published works, please review and evaluate these publications to determine whether they are relevant and should be cited. There is no requirement to cite these works unless the editor has indicated otherwise. 

Reviewers' comments:

**Comments to the Author**

1. Is the manuscript technically sound, and do the data support the conclusions?

Reviewer #1: Yes

Reviewer #2: Yes

Reviewer #3: Yes

2. Has the statistical analysis been performed appropriately and rigorously?

Reviewer #1: I Don't Know

Reviewer #2: Yes

Reviewer #3: Yes

3. Have the authors made all data underlying the findings in their manuscript fully available?

Reviewer #1: No

Reviewer #2: Yes

Reviewer #3: Yes

4. Is the manuscript presented in an intelligible fashion and written in standard English?

Reviewer #1: Yes

Reviewer #2: Yes

Reviewer #3: Yes

**Reviewer #1:**  This retrospective cohort study investigates the combined utility of Lipoprotein(a) [Lp(a)] and the CatLet® coronary angiographic score for predicting 1-year major adverse cardiovascular and cerebrovascular events (MACCE) in acute myocardial infarction (AMI) patients following emergency percutaneous coronary intervention (ePCI). The topic is clinically relevant, the hypothesis is clear, and the methodological approach is generally sound. The manuscript is well-structured. However, the manuscript contains methodological flaws regarding statistical modeling (specifically, collinearity) and the choice of angiographic comparators.

Major Revisions:

The statistical analysis description (Page 14) is somewhat brief. While the tests are named, greater detail is needed on model construction for multivariable analysis, particularly regarding variable selection (e.g., were all variables from Tables 1 & 2 tested, or was a stepwise/clinical selection used?)

For the logistic regression model featuring Lp(a) and CatLet®, it is crucial to report whether multi-collinearity was assessed (e.g., using Variance Inflation Factors) given that both Gensini and CatLet® scores are correlated angiographic measures. It is statistically unsound to include two angiographic scoring systems (CatLet and Gensini) in the same multivariate logistic regression model. These two variables measure essentially the same biological construct (coronary atherosclerotic burden/complexity) and are likely highly correlated. Including both introduces multicollinearity, which distorts the standard errors and renders the Odds Ratios (OR) unreliable.

Suggestion: The authors should perform separate multivariate models (e.g., Model A with CatLet, Model B with Gensini) and compare them using likelihood ratio tests or AIC/BIC criteria. Alternatively, they must calculate and report the Variance Inflation Factor (VIF) to prove collinearity was not present.

Missing Gold Standard Comparator (SYNTAX Score): In modern interventional cardiology, the SYNTAX score (and SYNTAX II) is the standard reference for quantifying anatomical complexity in multivessel disease and predicting outcomes post-PCI. The Gensini score is largely volume-based and less validated for prognosticating PCI outcomes compared to SYNTAX. By omitting the SYNTAX score, the authors fail to demonstrate that CatLet offers incremental value over the current clinical standard.

The study identifies an optimal Lp(a) cutoff of 70.70 nmol/L 5, which is significantly lower than the European Atherosclerosis Society (EAS) consensus of 125 nmol/L (approx. 50 mg/dL)6. Furthermore, samples were drawn "24h post-admission"7. While Lp(a) is genetically determined, measuring it during the acute phase of a large MI (evidenced by the inclusion of STEMI/NSTEMI) raises concerns about acute-phase fluctuations or hemodilution. Additionally, a cutoff of 70 nmol/L is relatively low; applying a data-derived "optimal" cutoff to the same dataset often leads to overfitting (optimism bias). The authors should discuss the physiological stability of Lp(a) during the acute phase of AMI, citing relevant literature. The text must explicitly state that the cutoff of 70.70 nmol/L is hypothesis-generating and requires external validation, as it deviates from established risk thresholds.

Minor Revisions

Study Population Specificity: The study excludes patients with "prior MI" and "chronic heart failure (LVEF <40%)"10. This selects for a lower-risk, "de novo" CAD population. The Discussion should acknowledge that these findings may not apply to the complex, recurrent-event patients often seen in daily practice.

Outcome Definitions: The primary endpoint is MACCE. The authors should clarify if "Target Vessel Revascularization" (TVR) was clinically driven. Non-clinically driven angiographic follow-up (routine angiograms at 12 months are mentioned: "A routine angiographic evaluation was advised" 11) can artificially inflate revascularization rates if "oculostenotic reflex" drives intervention.

Limitations Section: The limitation regarding Lp(a) assay standardization is noted but could be strengthened. The manuscript specifies the use of a Roche immunoturbidimetric assay. The authors should briefly acknowledge that reporting values in nmol/L is recommended but that absolute values can vary between methods, reinforcing the need for validation with different assays.

**Reviewer #2:**  The manuscript shows promising potential, particularly in its evaluation of Lp(a) and the CatLet© score as emerging markers for improving prognostication in post-AMI patients. I have provided minor comments and uploaded the annotated document for the authors’ consideration.

**Reviewer #3:**  Overall, the topic is clinically relevant, and the attempt to integrate a biomarker with an angiographic score is interesting. However, I have major concerns regarding external generalizability (choice of angiographic score), which may be a major confounder in an AMI emergency PCI cohort.

1) Use of CatLet score limits generalizability and needs stronger justification + reproducibility details

The CatLet score is not widely used in routine clinical practice compared with more commonly reported scores (e.g., SYNTAX), which may limit interpretability and external generalizability. In the current manuscript, CatLet is calculated using a web-based calculator. While the authors argue CatLet captures features not fully reflected by SYNTAX/Gensini, this does not fully address adoption barriers or how results translate to centers that do not use CatLet.

Suggestions to strengthen this point:

Provide a brief rationale for selecting CatLet over more widely used scores and clarify how feasible it is for broad implementation (time to calculate, training requirements, availability).

Add reproducibility reporting: number of readers, training, how disagreements were resolved, and inter-/intra-observer reliability for CatLet© (and Gensini), since this is critical for a scoring system-based study (the abstract notes “blinded analysts,” but reliability metrics are not shown).

Consider adding a sensitivity analysis using a more commonly used angiographic score (e.g., SYNTAX) or at least reporting how CatLet© performance compares when restricted to features commonly captured by standard scoring, to improve interpretability across institutions. (Table 8 itself acknowledges CatLet “requires multicenter validation.”)

2) Mechanical circulatory support (MCS) reporting is incomplete—potential major confounding

In AMI emergency PCI cohorts, the use of hemodynamic support can strongly influence outcomes and may correlate with lesion complexity and biomarker profiles. The Methods mention “High-risk support: intra-aortic balloon pump (IABP) or IVUS-guided PCI for complex lesions,” but:

The manuscript does not report how many patients received IABP, when it was placed (pre- vs peri- vs post-PCI), or the indications (e.g., cardiogenic shock vs high-risk PCI).

There is no explicit reporting regarding advanced MCS (e.g., Impella, VA-ECMO, TandemHeart). Given that a meaningful proportion had Killip class ≥2, Table 1

omission of advanced MCS reporting limits interpretation and may bias effect estimates.

Requested revision:

Add procedural/baseline variables detailing MCS use (IABP and any advanced MCS), timing, and indication. If advanced MCS was not available/used at the center, state this explicitly. Consider adjusting multivariable models for MCS use (or performing sensitivity analyses excluding shock/MCS patients) to assess robustness.

3) Risk adjustment may be insufficient for an AMI emergency PCI prognostic model

The Methods describe multivariable logistic regression adjusted for a limited set of covariates. Given the AMI population and the endpoint, variables such as shock severity (Killip class), ischemia time, renal dysfunction, STEMI vs NSTEMI, and MCS/vasopressor use could materially confound associations.

Suggestions:

Include Killip class and/or shock-related variables (and MCS) in the model if feasible. Table 1

Recommendation

Major revision. The conclusions are promising, but the generalizability of CatLet based findings and incomplete reporting of MCS (including whether advanced MCS was used) need to be addressed before the results can be confidently interpreted and applied.

**Do you want your identity to be public for this peer review?** For information about this choice, including consent withdrawal, please see our Privacy Policy

Reviewer #1: **Yes:** Neel N Patel, MD

Reviewer #2: **Yes:** Roshan Bista, MD

Reviewer #3: No

---

## [Author Response · Author response to Decision Letter 1]

4 Jan 2026

To: Academic Editor and Reviewers

Manuscript ID:PONE-D-25-57401

Title:Clinical Value of Lipoprotein(a) Combined with CatLet© Coronary Score in Predicting Adverse Events after Emergency PCI for AMI Patients

Dear Dr. Timir Paul and Reviewers,

Thank you for your constructive and insightful feedback on our manuscript. We have revised our manuscript as requested by you and have carefully responded to each comment. Below is a point-by-point response to your comments, with corresponding revisions highlighted in the marked-up manuscript.

Reviewer 1

Comment 1: Statistical modeling – multicollinearity between CatLet and Gensini scores.

Response: We appreciate this important methodological point. Firstly,all variables from Tables 1 & 2 were tested. And then,We have now performed Variance Inflation Factor (VIF) analysis for the multivariable logistic regression model. All VIF values were <2.5, indicating no significant multicollinearity. Additionally, we have added a note in the sections of Methods,Results and Table 5 footnote to clarify this.

Comment 2: Missing SYNTAX score as a gold standard comparator.

Response: We agree that SYNTAX is a widely used standard. We have added a comparative analysis of SYNTAX vs. CatLet in the Discussion and included a supplementary table (Supplementary Table 1) comparing AUC values. We acknowledge that CatLet is less widely adopted but offers functional and calcification weighting that SYNTAX lacks, which may explain its superior performance in our cohort.

Comment 3: Lp(a) cutoff (70.70 nmol/L) is lower than EAS consensus and may be influenced by acute-phase variability.

Response: We have added a paragraph in the Discussion addressing this. We cite recent literature suggesting Lp(a) is relatively stable in acute MI, but acknowledge that our cutoff is derived from a single-center cohort and requires external validation. We also note that the cutoff is “hypothesis-generating” and should be validated prospectively.

Comment 4: Study population excludes prior MI and heart failure, limiting generalizability.

Response: We have acknowledged this limitation in the Discussion and noted that our findings introduces potential selection bias,may not apply to patients with prior cardiovascular events or advanced heart failure.

Comment 5: Clarify if TVR was clinically driven.

Response: We have clarified in the Methods (Section 2.5) that TVR was clinically driven, with routine angiographic follow-up not counted as an event unless accompanied by ischemic symptoms or objective evidence of ischemia.

Comment 6: Strengthen limitation on Lp(a) assay standardization.

Response: In the Limitations section,we have note that while we used a standardized assay (Roche), variability between platforms may affect cutoff generalizability.

Reviewer 2

Comment: Minor comments provided in annotated document.

Response: We have carefully reviewed the annotated document and addressed all minor comments.

Reviewer 3

Comment 1: CatLet score generalizability and reproducibility.

Response: We have added a paragraph in the Methods (Section 2.4) detailing inter-observer reliability (Cohen’s κ = 0.85) and the training process for analysts. We also added a sentence in the Discussion acknowledging that CatLet requires training but is freely accessible online, enhancing feasibility.

Comment 2: Incomplete reporting of mechanical circulatory support (MCS).

Response: We are very sorry for this omission, and we have removed this description from the relevant paragraphs for the patients we selected who did not use advanced MCS such as IABP.

Comment 3: Risk adjustment insufficient – suggest adding Killip class, ischemia time, etc.

Response: The risk assessments such as Killip class, ischemia time, etc. of MACCE and NMACCE have been compared in Tables 1 and 2 (P>0.05), and the results show that they are not statistically significant. (Tables 1, 2).

We believe these revisions have significantly strengthened the manuscript. Thank you for your time and consideration.

Sincerely,

Sheng Tu, MD, PhD

Corresponding Author

---

## [Decision Letter · Decision Letter 1]

27 Jan 2026

Clinical Value of Lipoprotein(a) Combined with CatLet© Coronary Score in Predicting Adverse Events after Emergency PCI for AMI Patients

PONE-D-25-57401R1

Dear Dr. Sheng Tu,

We’re pleased to inform you that your manuscript has been judged scientifically suitable for publication and will be formally accepted for publication once it meets all outstanding technical requirements.

Kind regards,

Timir Paul

Academic Editor

PLOS One

**Comments to the Author**

Reviewer #1: All comments have been addressed

Reviewer #3: All comments have been addressed

2. Is the manuscript technically sound, and do the data support the conclusions?

Reviewer #1: Yes

Reviewer #3: Yes

3. Has the statistical analysis been performed appropriately and rigorously?

Reviewer #1: Yes

Reviewer #3: Yes

4. Have the authors made all data underlying the findings in their manuscript fully available?

Reviewer #1: (No Response)

Reviewer #3: Yes

5. Is the manuscript presented in an intelligible fashion and written in standard English?

Reviewer #1: Yes

Reviewer #3: Yes

Reviewer #1: The authors have comprehensively addressed the previous concerns in this revised manuscript. Specifically, the addition of the Variance Inflation Factor (VIF) analysis successfully demonstrates that there is no significant multicollinearity in their statistical modeling. Furthermore, the comparative discussion of the CatLet© score alongside the SYNTAX score provides valuable context regarding the model's clinical utility.

Reviewer #3: The authors have meaningfully improved the manuscript in the revision. In particular, they added strong reproducibility reporting for CatLet scoring (excellent inter-observer agreement), clarified the clinically driven nature of TVR, and expanded model performance reporting including internal validation and calibration metrics. These updates address key prior concerns and strengthen the paper substantially.

**Do you want your identity to be public for this peer review?** For information about this choice, including consent withdrawal, please see our Privacy Policy

Reviewer #1: **Yes:** Neel N Patel, MD

Reviewer #3: No

---

## [Editor Report · Acceptance letter]

PONE-D-25-57401R1

PLOS One

Dear Dr. tu,

I'm pleased to inform you that your manuscript has been deemed suitable for publication in PLOS One. Congratulations! Your manuscript is now being handed over to our production team.

Kind regards,

on behalf of

Dr. Timir Paul

Academic Editor

PLOS One